# Working Conditions and Mental Health in a Brazilian University

**DOI:** 10.3390/ijerph20021536

**Published:** 2023-01-14

**Authors:** Livia de Oliveira Borges, Georgina Maria Véras Motta, Geraldo Majela Garcia-Primo, Sabrina Cavalcanti Barros, Camila Teixeira Heleno

**Affiliations:** 1Programa de Pós-Graduação em Psicologia, Universidade Federal de Minas Gerais (UFMG), Belo Horizonte 31270-901, Brazil; 2Universidade Federal de Minas Gerais (UFMG), Belo Horizonte 31270-901, Brazil; 3Hospital das Clínicas, Universidade Federal de Minas Gerais (UFMG), Belo Horizonte 30130-100, Brazil; 4Faculdade de Ciências da Saúde do Trairi, Universidade Federal do Rio Grande do Norte (UFRN), Natal 59200-000, Brazil; 5Faculdade Interdisciplinar em Humanidades, Universidade Federal dos Vales do Jequitinhonha e Mucuri (UFVJM), Diamantina 39100-000, Brazil

**Keywords:** working conditions, precarious work, well-being, analysis of variance, social causation

## Abstract

The highest prevalence of mental illnesses and mental suffering in contemporary society has raised awareness of the theme and their connection to work. In Brazil, university servants (professors and technical-administrative staff) are a focused occupational group. We developed this research with the objective of exploring the relationship between the perception of working conditions and the mental health of these servants. Structured questionnaires were applied to 285 servants, 33.5% being professors and 66.5% technical-administrative staff. Regarding working conditions, the questionnaires included items that measured 15 primary factors and questions about their contracts and legal conditions. To evaluate mental health, the participants answered a questionnaire about common psychic symptoms, negative and positive affects, self-esteem, and family-work conflict. We composed groups of participants according to their mental health indicator scores (cluster analysis), and after that, we compared the mean scores in working conditions for the groups. Then, we found that the mean scores of 13 from the 15 working condition factors were significantly different between the mental health groups. Our results showed the importance of improving working conditions in universities to prevent mental illnesses. Understanding the content of each working condition factor presents potency to contribute to defining the priorities among different aspects of working conditions.

## 1. Introduction

The highest prevalence of mental illnesses and mental suffering in contemporary society has put the theme and the connection between work (and/or its conditions) and mental health in the spotlight [1,2,3]. In Brazil, among the most focused occupational groups are public servants and, among these, university workers, such as professors or technical-administrative staff (TAS) [1,4]. The Brazilian context of university civil servants has also been problematized due to institutional transformations in the face of the devaluation of education and science by public policies in the country. Among such transformations, we observed: the reduction and volatility of public funds for the maintenance of organizations, research activities, and services [5]; outdated salaries, changes in social security policies, and retirement funds [1,6]; demands that surpass the possibilities provided for in the employment contract in terms of working hours [1]; the increasing and simultaneous migration of such bureaucratic processes to digital portals [7]; the loss of autonomy with the standardization of activities; the transferring of administrative-bureaucratic attributions to professors and less TAS being hired [4,8]; and increasing pressure for a higher academic production that does not always consider the quality of what is done [9,10]. These transformations have resulted in the imbalance between institutional and competence requirements, the increasing overload, and the loss of identification of servants with their activities and/or charges. Based on a literature review, Costa and Souza Neto [1] pointed out, among other aspects, the tendency of a directly proportional relationship between the number of scientific productions and students advised per year with the occurrences of cardiac procedures, coronary problems, and strokes in university professors. Some of these problems are specific to Brazilian reality; however unsatisfactory working conditions are a more general problem in today’s world [11].

This whole situation led us to the development of this research with the objective of exploring the relationship between the perception of working conditions and the mental health of public servants in the Universidade Federal de Minas Gerais (UFMG, Federal University of Minas Gerais). We hope to have contributed to the understanding of the mentioned links, as well as being useful to Brazilian universities to support the establishment of prevention strategies in mental health.

### 1.1. Concepts of Mental Health, Assumptions, Indicators, and Main Theoretical Models

The field of research and professional activity in mental health and work has maintained the tension on the recognition or not of the connection between work and mental health. It means a fundamental contradiction, in which it is necessary to recognize this connection to make sense of the existence of the field, and simultaneously, part of the research tends to focus on themes, problems, and hypotheses to demonstrate such connection [12]. This recognition, in turn, requires assuming work as a structuring social category in the lives of people and societies [13,14,15].

Going through the history of the aforementioned field, we begin by noting that Le Guillant [16], a psychiatrist, had already theorized at the end of the 1950s about the importance of economic, social, political, and ideological contexts in shaping working and living conditions and described how characteristics of work organization and management affect the development of psychological symptoms. He started with a case study of his clients. Among his studies, the case of the neurosis of telephone operators was well known, in which the workers had nervous fatigue associated with repetitive and fast work. For Le Guillant [16], technology and work organization reduced muscle effort but accelerated the pace. He defended that the occurrences of illnesses are not equally distributed among the different occupations of workers. Thus, in reporting the case of the Papin sisters, he underlined: “In 278 suicide attempts (among women) at the Necker Hospital (...), 85.34% refer to housekeepers, maids, cleaners, and saleswomen (at food shops, these have practically the same status); only 3.6% of them are typists; 3.9% technicians; 3.6% factory workers; 2.9% students; and 0.70 ‘artists’”(pp. 325–326). In other words, he showed the relationship between the prevalence of psychic alterations by occupational groups. Therefore, his contributions moved towards the construction of a psychopathology of work, as well as towards a social and epidemiological perspective. Thus, if work is a structuring social category, working conditions can promote and/or deteriorate health. Le Guillant [16] then called attention to the need to elucidate the work conditions that promote and deteriorate health, as well as to identify endemic illnesses by occupation. Several studies continue to be carried out in this direction [3,17,18,19,20,21,22].

The aforementioned contributions by Le Guillant [16] were influential in building the field of mental health and work. However, there were several other notable influences, such as the positive concept of health by the World Health Organization [23], which consisted of taking health as a synonym for a complete state of well-being and not just the absence of illness. More recently, the WHO updated this concept with a view to better application for the benefit of public policies. Thus, he abandoned the notion of fullness and defined that being healthy means developing sufficient autonomy to lead a productive social and economic life [24]. The incentive to adopt the positive concept was expressed in initiatives such as Goldberg’s [25,26] to pay attention to common (or minor) psychic alterations, both due to the feasibility, at the time, of identifying among active workers and/or employees, as well as the potency to support preventive actions.

In such a theoretical context, Warr [27] proposed the ecological model of mental health. He expressed the positive concept of health, defending its multidimensionality and adopting five health indicators, namely: affective well-being, integrated functioning (which encompasses the balanced development of different spheres of life), personal competence, autonomy, and aspirations. He considered that the last three indicators could be summarized by self-esteem. He then postulated that health would vary according to nine environmental aspects, namely: the opportunity to control the environment, knowledge, and skills, variety of duties (or tasks), environmental clarity, money, perceived physical security, opportunity to develop interpersonal relationships, valuing social positions, and having externally generated goals. The variances between such environmental factors and health tend to be linear in the lower ranges of scores, but after a certain point, they tend to present two distinct lines: (1) forming a plateau, known as a constant effect, indicating that the increase in the environmental factor, after a certain point, is no longer followed by improved health, as is the case with aspects related to money availability, perceived physical security, the opportunity to develop interpersonal relationships and valued social position; and, (2) forming a curve in a downward facing parabola, called additional effect, in which the increase in environmental factors is followed by a growing deterioration in mental health, as is the case with aspects related to the opportunity to control the environment, the development of knowledge and skills, having externally generated goals and environmental clarity. Therefore, shared variances are complex. We observed that all these aspects could be applied in the social environment as a whole, as well as in the work environment. Thus, we understand that Warr [27] offered a new direction for the exploration of endemic illnesses in occupations and organizations, not needing to consider only the diagnoses of illness conditions.

In attempts to demonstrate the link between mental health and work, studies have also used the strategy of comparing the mental health of employed and unemployed people [28]. The books by Mirowsky and Ross [29,30], however, reported longitudinal research in which the authors constructed groups by social status using a set of sociodemographic indicators (e.g., income, educational level, gender, race, marital status, etc.). They understood that social position contributed to people’s perception of being in control of their lives, which is fundamental for mental health. They found variations in mental health by social position and that this relationship was moderated by macrosocial characteristics such as the level of inequality in society, the contribution of society for its citizens to maintain or overcome alienation, and the degree of legitimized authoritarianism. This model was called social causation and placed work at the origin of health promotion/deterioration along with other dimensions of social life (family). As for the authors, control of life (in the sense of resolving everyday problems) was central to health promotion; from the second edition of the book Social Causes of Psychological Distress, they highlighted the role of education in the development of life control.

In the same context of the influence of the positive concept of health, the models related to stress grew, with emphasis among them on the model of control and demands by Karasek [31]. He understands that stress (not necessarily pathological) develops in the contradictions experienced in the work environment between the possibilities of controlling the environment and the demands that are presented to workers in the management and organization of their work. Johnson and Hall [32] complemented this model, showing the moderation of social support in the relationship between the two aspects mentioned above. The model was tested in longitudinal studies as reported in publications such as those by Jonge, Vegchel, Shimazu, Schaufeli, and Dormann [33] and Hart [34]. At the same time, there was recognition of burnout syndrome, originating from workplace situations and prevalent in human care professions and services. It is a set of symptoms originating from experiencing chronic stress in the work environment. On the subject, the psychosocial approach, with Maslasch and Jackson [35] among its initial drivers, was the approach that most flourished [20], including the studies developed in the academic environment as in the recent papers by Lackritz [36] and by Alves, Oliveira, and Paro [37].

Both the stress models and the research on burnout syndrome were in line with the notions of social causation and the idea of the prevalence of alterations by occupations. However, for our research, we opted for the identification of common or minor psychic alterations and the positive mental health indicators, which we will refer to again in the Section 2. We considered them more comprehensive in the sense of closely representing the WHO concept of health proximately. In other terms, our intention was to focalize mental health as a whole and not only a specific type of alteration.

Other authors, such as Martín [38], have been concerned with understanding what an environment poor in well-being is and exploring the relationships with mental health symptoms. However, all of this is challenging given the complexity of the world of work, which is continuously changing. Therefore, we currently demand models that meet the diversity of the world of work in the IV Industrial Revolution. This phase of capitalism has been described as encompassing four dimensions: (1) technological (artificial intelligence, hyperconnectivity, integration of the physical, biological, and digital worlds, genetic sequencing, metadata, among others): (2) organizational and/or managerial (generalization of the use of networks, closer ties between producers and consumers, expansion of the diversity of services provided, web economy, etc.); (3) socioeconomic (changes in contract and legal labor relations, resizing of occupations and professions, cost reduction policies, etc.); and, (4) political-ideological that increases the emphasis on values such as individualism, competitiveness, and individual competence [39,40,41].

The specialized literature has proposed models that enable researchers and professionals to apprehend the ongoing changes. One of these models was proposed by Marchand and Durand [42], with the designation of biopsychosocial. It starts from the social origin of the symptoms, operating in the experiences of the subjects in four spheres: work, family, network, and community. From such experiences, subjects experience stress with its psychosomatic consequences. Consequently, they could present symptoms such as chronic psychological stress, depression, burnout syndrome, and abuse of alcohol and other substances, among others. The development of symptoms would depend on the types of moderators that affect the subjects’ experiences in the four spheres and the experiences of stress. The moderators referring to the subject’s social insertion are gender, life cycles, psychological traits, and stressful life events. The societal moderators, in turn, consist of unionization, size of organizations, economic sector, and labor market instability.

In summary, our research was planned considering that mental health is a multidimensional phenomenon, demanding indicators of psychological alterations and positive characteristics of individuals (affective well-being, integrated functions, and self-esteem). A healthy person presents autonomy and control of his or her life and is able to cope with their symptoms or alterations as well as the challenge of the environment. Then, mental health is a dynamic phenomenon which people build in their social insertion and not a phenomenon exclusively from people’s interiority. Social insertion can be thought of at various levels of analysis, such as interpersonal, organizational, collective, and societal levels.

Transformations in the world of work also require a reinterpretation of factors in the work environment in light of societal changes, especially regarding labor policies. So, for the present research, we considered the need to better operationalize people’s experiences with their work. It is true that Warr’s [27] environmental factors are useful for that, as well as the moderators could be better operationalized considering the societal aspects addressed in Mirowsky and Ross’s model of social causation [29]. However, we considered there to be a gap in developing this approach in the sense of covering the working conditions in conformity to cited transformations. Moreover, we found a more detailed systematization that better covers work today that refers to working conditions, which we will deal with in the subsequent section.

### 1.2. Working Conditions

The conception of working conditions adopted here is based on two main inspirations. The first is the concept of decent work, namely:

Decent work means productive work in which rights are protected, which generates an adequate income, with adequate social protection. It also means sufficient work, in the sense that all should have full access to income-earning opportunities. It marks the high road to economic and social development, a road in which employment, income and social protection can be achieved without compromising workers’ rights and social standards. [43] (p. 6).

This concept demands, in turn, a comprehensive definition of working conditions that involve the environment, content (including work organization), and psychosocial aspects [44,45,46,47,48]. The second inspiration is the research carried out by the European Foundation for the Improvement of Living and Work Conditions (Eurofound); a tripartite agency created in 1975 to subsidize the development of public policies aimed at improving living and working conditions [44,49]. Cardoso and Morgado [46] highlighted that this agency has conducted surveys every five years since 1990, addressing the following dimensions: professional context; working hours; work intensity; physical, cognitive, and psychosocial factors; welfare; competencies, qualifications, and career prospects; work organization; social relationships; professional achievements; violence, harassment, and discrimination; reconciliation of professional life and family life; and financial security.

In Brazil, these antecedents were organized into a taxonomic synthesis that deals with the various components of working conditions organized into four categories: (1) contract and legal conditions, referring to the types and modalities of work and contract modalities; (2) physical and material conditions, concerning the more concrete aspects of the work environment; (3) processes and characteristics of the activity, focused on aspects of the content of the activities; and (4) conditions of the socio-managerial environment, referring to interpersonal interactions and social practices, within organizations and/or other insertions in the labor market [50]. In Brazil, the reality of occupations is quite different. Therefore, Borges et al. [50] developed a specific questionnaire on working conditions for university employees (professors and technical-administrative staff). The first category (contract and legal conditions) is clearly composed of nominal indicators such as type of contract (Consolidation of Labor Laws [individuals under standard employment relationships], statutory, self-employed, individual micro-entrepreneurs, digital platform workers, etc.), working hours, employment contracts (temporary, for unspecified lengths of time, etc.). The other categories include indicators to which a standardized frequency Likert-type scale (from Never to All the Time) can be applied. Therefore, they were the object of empirical exploration with the application of factor analysis. Exploratory factor analysis resulted in the identification of the factor structure synthesized in Figure 1.

Thus, in the present research, the intention was to partially apply the model by Marchand and Durand [42], exploring only the variance in the mental health of employees of a federal public university according to work experiences, operationalized by the perception of working conditions according to the primary factors (Figure 1). Each factor is conceptualized in Figure 1. This factorial structure represented a new manner of apprehension of the current working reality in the university context. We also explored the moderation of shared variance by aspects of contract and legal conditions (professors versus technical-administrative employees, servants in the probationary stage versus the others). The probationary period is the evaluation period of the civil servant before they become permanent. It corresponds to the first three years of his or her career, during which the server is still being evaluated and does not have tenure.

## 2. Materials and Methods

We developed this research at the Universidade Federal de Minas Gerais (UFMG, Federal University of Minas Gerais). This public institution is one of the most renowned better universities in the Brazilian system. It is installed on four campuses: Pampulha, Saúde, Montes Claros (Regional Campus), and Tiradentes (Cultural Campus). The UFMG offers 91 undergraduate courses, 81 postgraduate programs (with master’s and doctorate levels), and 21 master’s-only programs. Furthermore, the university also offers specialization courses (lato sensu), as well as other short-term complementary courses. The students do not pay anything to enroll in undergraduate and graduate courses. The UFMG employs 7414 people as effective servants in two occupational groups: 3200 professors (43.16% of the total) and 4214 TAS (56.83% of the total). The former acts in the university functions, which are lecturing, researching, services, and management. The latter composes a group that covers all other staff, including bureaucratic and administrative activities that do not require a college degree and technical activities that require a college degree (for example, engineers, psychologists, doctors, librarians, etc.). These occupational groups are differentiated and inclusive in their unionization: each one has its own union. (https://ufmg.br/a-universidade/apresentacao/ufmg-em-numeros, accessed on 30 December 2022). In Brazil, federal universities have experienced various changes and coping problems. The considerations already presented at the beginning of this text can be applied to UFMG’s reality, as well as other federal universities.

### 2.1. Participants

A total of 285 civil servants from the UFMG participated in this research, 33.5% of them professors and 66.5% TAS. They presented a mean age of 41.6 years (SD = 10.9). There were 175 (61.6%) women and 109 (38.4%) men. In reference to the educational level of the participants, we found that: four (1.5%) of them had completed high school; 51 (17.9%) had incomplete or complete higher education; 93 (32.7%) had a lato-sensu postgraduate (specialization); 48 (16.9%) had a Master’s; 88 (31%) had a doctorate course (graduation). Professors at UFMG are currently hired with the requirement that they have a graduate degree. However, it is still possible to have professors without this qualification level as a result of the shortage of qualified people in some academic areas. The number of graduated participants also included five TAS who have this title, regardless of whether it is a requirement for the positions held.

Regarding marital status, 94 (33.1%) participants were single, 161 (56.7%) were married or lived with a partner, and 29 (10.2%) were divorced, widowed, and/or unidentified. Regarding the salary range, 90.8% of the participants received salaries above BRL 3000.00 and below BRL 20,000.00. Finally, it can be observed that for 87.% of the participants, their jobs at UFMG are their only source of income. Of the sample, 188 (66.2%) are not unionized. Of the 96 unionized workers (33.8%), 51.% stated that they sometimes participated, often, or continuously in union activities.

### 2.2. Field Activities

Field activities took place in the second half of 2017, therefore, before the period of the COVID-19 pandemic. Such activities consisted of the application of digital questionnaires (online). Aiming at the security of responses in terms of quality of registration, secrecy, and anonymity, a computer service prepared the digital format of the questionnaires using the JavaScript Object Notation (JSON) data structure. This format of the questionnaires prevented the completion of the form, skipping questions, and ensuring the participant the possibility of suspending the completion of the form and resuming it at another time. Before completing the questionnaires, each participant had access to the message about the research objectives and the voluntary and anonymous nature of participation.

For these procedures, the questionnaires were sent to all servants. However, voluntary participation resulted in an accidental sample, in which inclusion criteria were the participants’ decisions. This voluntary nature of participation also resulted in a sample in which TAS are represented in a larger proportion than in the population. However, it is important to register that they represent the majority occupational group.

### 2.3. Instruments

To apprehend the perception of the working conditions of the participating servants, we applied the Working Conditions Questionnaire (WCQ) [48]. The WCQ is a version specially prepared for university servants and has an appreciation for its validity and consistency. It measures 15 primary factors organized into three secondary factors (Figure 1), in addition to having items on contract and legal working conditions. Responses to these items and the sociodemographic form (gender, educational level, etc.) were used for the description of participants in the previous section.

To measure mental health indicators, we used the following questionnaires, in their Portuguese-language versions: General Health Questionnaire (GHQ-12) [26,52,53], Job-Related Affective Well-Being Scales (JAWS) [54], Rosenberg’s Self-Esteem Scale (RSS) [55], Work-Family Conflict Scale (WFCS) [56]. We chose these questionnaires with the intention that the first questionnaire would allow us to identify common psychological symptoms. We consider that an indicator of this nature is in line with Marchand and Durand [42] and Martín [38]. Taking inspiration from the WHO’s positive concept of health, as we mentioned earlier, should not mean denying the psychic changes already experienced by workers. This differs, however, from Le Guillant [16] because it does not replace the diagnosis of a nosologically recognized illness. The GHQ-12 is understood as a probing measure [53]. The adoption of JAWS complements the GHQ measure with affective content so that we can measure Warr’s Affective Well-Being dimension [27] in a bipolar way. The RSS was included considering Warr’s point of view [27] that self-esteem synthesizes competence, autonomy, and aspiration. The choice of the WFCS had the objective of apprehending, at least partially, the dimension of Warr [27] of integrated functioning. Therefore, the multidimensionality of mental health supported the methodological planning of the research.

All adopted questionnaires relied on previous research on the evidence of their validity and consistency in Brazil. However, we re-examined them all for the specific sample of UFMG servants, with a view to greater reliability of the results and being closer to the reality of the public university.

Thus, we submitted the RSS responses to confirmatory factor analysis (CFA) and only obtained a solution with adequate adherence indices with the elimination of half of the items (1, 2, 3, 4, and 7). GFI (Goodness of Fit Index) = 0.989; AGFI (Adjusted Goodness of Fit Index) = 0.967; PGFI (Parsimony Goodness of Fit Index) = 0.30; CFI (Comparative Fit Index) = 0.995; IFI (Incremental Fit Index) = 0.995; RFI (Relative Fit Index) = 0.973; RMSEA = 0.043, χ2 = 7.59 para *p* = 0.8. We tried a bifactorial solution, but it proved to be unfeasible because it does not obtain discriminant validity. We were then left with the one-factor solution. Regarding JAWS, we started from the factorial solution available in the literature but estimated Cronbach’s alpha coefficients that resulted in α = 0.93 for Positive Affects and α = 0.88 for Negative Affects. Regarding the responses to the GHQ-12, we followed the guidelines in the literature [53] to consider it unifactorial. However, we submitted it again to a CFA to evaluate the adequacy of both the unifactorial solution and the characteristics of the present sample of university employees. We found good fit indicators when we subtracted some items with factor loadings below 0.60, leaving seven items (α = 0.90). GFI (Goodness of Fit Index) = 0.931; AGFI (Adjusted Goodness of Fit Index) = 0.862; PGFI (Parsimony Goodness of Fit Index) = 0.466; CFI (Comparative Fit Index) = 0.949; IFI (Incremental Fit Index) = 0.949; RFI (Relative Fit Index) = 0.906. We highlight that a desirable PGFI is ≤0.50 and the others >0.90.

Finally, we repeated the CFA for the WFCS and found good adherence coefficients after eliminating items 2, 3, and 6. GFI (Goodness of Fit Index) = 0.976; AGFI (Adjusted Goodness of Fit Index) = 0.949; PGFI (Parsimony Goodness of Fit Index) = 0.453; CFI (Comparative Fit Index) = 0.991; IFI (Incremental Fit Index) = 0.991; RFI (Relative Fit Index) = 0.968; RMSEA = 0.054, χ2 = 23.94 to *p* = 0.03. In addition to Lisrel’s own indications in this direction, we found that these items created multicollinearity among themselves and with other items. However, factor loading errors persisted (greater than 1.0). We estimated, on the other hand, Cronbach’s Alpha coefficients and found α = 0.88 for both factors (Work to Family Interference [WFI] and Family to Work Interference [FWI]).

### 2.4. Response Analysis Procedures

We estimated WCQ primary factor scores by the load-weighted means of the items [50]. For the factors measured by the other structured questionnaires, we estimated the scores using the arithmetic mean of the points assigned to each item. To the scores on the mental health factors, we applied a cluster analysis that allowed us to identify mental health profiles shared by sample groups. Finally, with the objective of raising the variance of the scores in the factors of working conditions by mental health profiles, we applied the ANOVA method.

## 3. Results

### 3.1. Working Conditions

After estimating the participants’ scores on the working condition factors (Table 1), we found consistency with those of a broader sample [57]. In this way, the results also highlight the perception of working conditions guarding experiences of contradictions, as in the category of Demands, Contradictions, and Material Conditions, the highest scores are precisely in the Respect for Limitations and Material Precariousness primary factors. Respect for Limitations covers the perception of acceptance of the servant’s right to carry out their health treatments, to leave occasionally, to take breaks, and other limitations. Material Precariousness concerns the perceived insufficiency of available equipment, materials, and software. Then, we observe that the third mean in magnitude occurred in Demands, which encompasses the visibility of performance charges, as well as the imbalance between such charges and the available infrastructure.

The mean scores in the Bureaucratic process and Violence primary factors were close to the midpoint of the scale (scale from 1 to 5, with a midpoint of 2.5), and in the first of these factors, it is noteworthy that the last quartile of the distribution was 3.15, meaning that 25% of the sample scored above this point on that factor. Therefore, the proportion of participants who are uncomfortable with the bureaucratization of the university is not small. We must remember that this factor includes ignorance of bureaucratic rationality. It is also noteworthy that the last quartile of the distribution of scores in the Violence factor begins at point 2.84. We highlight that the Violence factor refers to the perception of moral harassment, intimidation, verbal aggression, and threats, among others. The perception of violence in working conditions tends to involve difficulties in recognizing it [58], which leads us to consider the starting point of the last quartile of the distribution as a warning sign. However, Hassard, Teoh, and Cox [59], through a systematic review, warned us not to forget the economic costs (for organizations and for society) likely to occur as consequences of the sequelae of experiences of violence. Hassard et al. [59] identified productivity losses, absences, and presenteeism among organizational costs. To society, he identified intangible (pain, suffering, and reduced quality of life), indirect (productivity losses), and direct costs (medical care expenditures).

Among the distributions of scores in the factors related to Communication, Challenge, and Participation, it is notable that the highest mean is in Computerization, which consists of the perception of the use of information technology and media. The distribution of scores in this factor indicates that this process is an important configurator of the perception of working conditions in the opportunity of field activities. In addition, the other distributions showed that the perception of the participants revealed strong aspects of the University. Probably, the opportunities to update knowledge, participate in collective decisions, as well as the sense of responsibility and value the identity of the servants towards the academy. The distribution of scores that differs from this tendency by presenting a lower mean refers to the Flexibility of Working Hours. That is the reason we registered that field activities took place shortly after the adoption of a clocking system to control the punch-in and out times of TAS.

In the second-order factor, Support and Planning, the scores of two factors (Standardization and Management by Objectives) also indicated that the participants tended to consider the covered aspects as strengths of the institution. Standardization refers to the perception of norms and guiding standards for carrying out the prescribed work. Management by Objectives, in turn, concerns the perception of how much the institution is guided by objectives in a climate of dialogue and conflict management. However, the scores in the Support and Protection factor revealed a tendency to perceive working conditions at UFMG as helpless, in other words, a tendency not to perceive sufficient actions to prevent illness and accidents, as well as little support from managers for the execution of services. This tendency is typical of the current precarious situation of working conditions.

### 3.2. Mental Health

Regarding mental health, the participants presented the scores summarized in Table 2. Half of the sample presented common psychic alterations (measured by the GHQ-12) with scores from 2.0 onwards, which should be understood as a warning sign [60,61]. Regarding affects, there is a tendency towards a moderate prevalence of positive affects over negative ones, as well as in both affects (positive and negative); most scores approached the midpoint of the scale (from 1 to 5). We also found a significant difference (t = 2.79; *p* ≤ 0.006) in positive affects, in which the mean score for professors (mean = 3.11; SD = 0.90) is higher than for TAS (mean = 2.79; SD = 0.90). Self-esteem scores showed a similar distribution, which is not desirable. There is a strong tendency to perceive work as interfering with family life more than the other way around. Additionally, we found a higher mean for professors (mean = 4.36; SD = 1.26) than for TAS (mean = 3.08; SD = 1.25), which is a significant difference (t = 7.9; *p* ≤ 0.001).

Valuing the notion of multidimensionality as discussed in the introduction, we grouped the measures through cluster analysis, identifying six health profiles as described in Figure 2.

The distribution of the health profiles is not independent of the occupational group of the servants (professors and TAS), with the Chi-square of cross distributions being significant (χ^2^ = 50.18; *p* ≤ 0.001). The profiles we call Indifferent Positives, Lively Positives, and With Intense Common Disorders prevail among TAS, and the opposite occurred with the profile entitled Self-valued Negativists.

### 3.3. Health Profiles according to the Perception of Working Conditions

We applied ANOVA to the scores of the working condition factors by health profiles and found (Table 3) that almost all first-order factors (13 out of 15) are capable of differentiating the mental health profiles (*p* ≤ 0.01), as the ANOVA result is statistically significant.

To explore whether contract and legal conditions (professors versus TAS, probationary public servants versus others) could moderate the relationships between other aspects of working conditions, we repeated the analysis of variance only for TAS. We found that the significance level for F2 Responsibility (Communication, Challenge, and Participation) was not maintained, but all others were maintained. We then repeated the same analyses only for professors and found that statistically significant relationships remained: (1) in the seven factors of Demands, Contradictions, and Material Conditions; (2) only in F3 Participation among the five factors of Communication, Challenge, and Participation; and (3) in F1 Support and Protection and F3 Management by Objectives, which are primary factors of the secondary factor Support and Planning. Then, there are different working conditions affecting the mental health of professors and TAS.

The moderation of being in the probationary stage (another component of contract and legal conditions) was tested by repeating the ANOVA only for new hires and then only for veterans. For new hires, only the results referring to the Responsibility factor (which ceased to be significant, F = 1.65 to *p* = 0.15) and the Standardization factor, which became significant (F = 2.56; *p* = 0.03). For veterans, there are variations in the F coefficients, but the results are still statistically significant in the same factors.

## 4. Discussion

In relation to the perception of working conditions, in summary, the results showed that the participants of the present research tend to perceive the working conditions at the university as adequate, with regard to the possibilities of updating knowledge, participation in decision-making processes (collective), for valuing the sense of academic responsibility, for respecting the health limitations of the servant, for advancing in computerization, for standardization, and for management by objectives. In contrast, they consider the working conditions to be inadequate regarding the impairment of performance requirements in view of the available infrastructure, the level of bureaucratization, material precariousness, persistent practices of violence, and insufficient institutional support and protection for servants. The positive aspects tend to strengthen mental health, while the negative ones weaken it since 13 out of 15 working condition factors were able to differentiate health profiles, as the results showed. In part, this occurred because the elaboration of the questionnaire was carried out considering the available knowledge about working conditions in public university institutions and the team’s experience with the university itself [48]. The scores that exposed deficiencies and contradictions in working conditions systematize previous findings in addition to corroborating them. Thus, the impairment found between performance demands on the one hand and available infrastructure and material precariousness on the other can be understood as another way of operationalizing the contradiction between demands and control of the environment, which Karesek [31] pointed out as the basis of stress development. This same contradiction, when persistent, has been considered a contributing cause of the development of burnout syndrome [36,37,62]. Such impairment also apprehends the effects of public policies that devalue education and science but maintain and/or expand the requirements for those who work in the area [1,5]. The perception of advances in computerization, followed by discomfort with the level of bureaucratization, represents what Oliveira et al. [4] indicated as a tendency to change the format of bureaucratic-administrative activities and transfer them to professors. While the attributions of professors are distorted, partial emptying of TAS activities is being promoted, justifying a reduction in hiring. The lack of institutional support and protection measured in the present study in relation to organizational management is probably a reflection, at least partially, of changes in social security policies and salary freezes [1,6]. In addition, it should be noted that previous studies, one on professor tutoring [63] and another on the removal of TAS in the probationary stage, also indicated feelings of helplessness among civil servants.

The results regarding mental health showed that, in the sample as a whole, there is, fortunately, a tendency for positive affects to prevail over negative ones, although such prevalence is less evident among the TAS. This difference requires inquiring about the reasons that motivate them. Are the TAS more negative in relation to the situation of universities? This question may stimulate future research. However, it is important to remember that Tessarini and Saltorato [64] have already shown that they tend to perceive themselves as inferior servants in their daily work. Probably, this perception reflects the interpersonal relationships in the university’s daily life and the (de)valuation of their activities in society as a whole.

Regarding the other indicators adopted, we must consider that the above-desirable proportion of participants with common psychic disorders, with lower-than-desirable self-esteem, and who perceive work as interfering with family life more than the other way around are warning signs. The tendency towards low self-esteem may be associated with the national context of the devaluation of science and education [1,5,6]. Regarding the interference of work in family and vice versa, we remember that we understand it as an operational indicator of integral functioning adopted by Warr [27] as one of the indicators of mental health. The strongest tendency among professors to perceive work as interfering with family more than the other way around probably reflects the daily life in which, in practice, there is no delimitation of the professor’s working hours. It is up to us to ask whether this undesirable tendency has not been worsened by the escalation of virtual work, encouraged by the experience of the COVID-19 pandemic, but typical of current tendencies in the world of work [39,40,41]. Leite [65] found, in a study where he interviewed professors, not only reports indicating that professors were working on dates and times that should be free and dedicated to other spheres of life but also showed a tendency to naturalize the extension of working hours justified by the need to publish and/or produce academically. Their interviewees did not report any action to face this situation. Leite’s publication took place in the same year that the field activities of this research took place. Ferreira and Oliveira [9] more recently mentioned the pressure for academic production and the intensification of work associated with extending the actual length of the working day.

Regarding the set of tendencies identified in the results referring to mental health, we must also investigate the relationship between these results and those found by Costa and Souza Neto [1], indicating a directly proportional relationship between the number of scientific productions and student guidance per year with occurrences of cardiac procedures, coronary problems, and cerebrovascular accidents. This question can be a starting point for future research in the direction of showing several endemic tendencies among university servants that are probably not dissociated from each other.

In order to have an understanding that is not limited only to the tendency of the distributions of the scores in each health indicator separately, we identified the mental health profiles of the participants using the technique of cluster analysis. This was a strong aspect of the research in an attempt to find a way to not be tied to studying illnesses but to the real health situation (with aspects of health and illness) of groups of participants. This path also has the merit of avoiding the psychiatric labels that, according to Dellwing and Harbusch [66], tend to be associated with the notion of medicalization and the tendency to transform people into problems (by stigmatizing them). On the contrary, the intention was to illuminate the notion of social causation in the present research.

The identified profiles showed that according to six health indicators, it is possible to draw several real profiles that are quite different from each other. They implicitly illustrated the likelihood that working conditions impact people and groups in different ways.

As previously shown, the scores for 13 working conditions factors (among 15 factors) differentiated these profiles, as well as contract and legal conditions (professors versus TAS, new hires versus veterans), and moderated this differentiating ability. We consider, then, that these results represent an adequate operationalization of the notion of social causality to the process of health promotion and common psychic alterations; we mentioned this notion in the introduction of this article.

If we reflect on each working condition factor, whose scores vary according to health profiles, and compare these results to those found in the literature, we can better perceive their value. Thus, among those grouped into Demands, Contradictions, and Material Conditions, we have already commented that the impairment between Demands, Material Precariousness, and Physical Environment can signal stress, following the model of demands and control of the environment. So, we understand that the shared variance of these factors and health profiles corroborate previous studies that point to the role of stress preceding mental illness [38,42,67]. It also corroborates the findings, in Brazil, by Costa and Souza Neto [1], mentioned in the introduction, who observed a tendency towards an imbalance between competence requirements, work overload, and the identification of the public servant with their activities. The significant shared variance of the Discrimination (as a form of violence) and Violence factors with health profiles corroborates the reviews that have pointed to the relationship of these phenomena with the development of post-traumatic stress and psychological illnesses [56,68]. The significant shared variance of the Respect for Limitations factor to the mental health profiles indicates that its maintenance can be a protective factor for the health of the servants. Therefore, they can reduce costs directly related to health care—this predictive factor may stimulate further studies. Finally, the shared variance of the Bureaucratic Process factor to the profiles corroborated the impacts of the tendency of growth of these processes and their migration to digital platforms already introduced [4,7,8].

Among the Communication, Challenge, and Participation factors, the significant variance of four of the factors (Knowledge Update, Responsibility, Participation, and Flexibility of Working hours) corroborate the importance of autonomous spaces for health promotion [27]. In addition, the reduction of autonomy in the university environment has already been pointed out in Brazil by other authors [4,8]. Among the Support and Planning factors, we point out that the variance of the profiles by the Support and Protection factor, as well as the tendency to lower scores in this factor, makes it urgent to remember that institutional support is a health-protective factor and, when specifically necessary, in cases of post-traumatic stress coping [69].

We also understand that working conditions, as measured in this research, apprehend aspects that concern the organizational management of universities, but without separating it from public policies (especially science and education) and societal tendencies in the world of work. Consequently, our understanding is that our results do not strengthen the recent approach regarding the increasing number of illnesses in servants caused by contagion among them due to high mobility, which was recently defended by Kensbock et al. [2]. This approach has already produced reactions in the academy. Thus, Pierce and Rider [70] developed an analysis drawing attention, among other aspects, to distinguishing contagion from normalization and the need to expand support to workers and/or employees. We consider here that the endemic symptoms of mental illness have increased, most likely due to the insertion of workers as a whole in precarious working conditions, in which public university servants in Brazil are a case, although aggravated by local public policies.

The variance of scores for the perception of working conditions by mental health profiles means that working conditions are a powerful phenomenon for mental health promotion and prevention (as an indicator of quality of life). It indicates that psychic health prevention at UFMG needs to include changes in working conditions and not simply use individual follow-up strategies in the face of early symptoms. Intervention after the onset of symptoms, even if it is understood as a level of prevention, is more strategies for treating an illness that is already in progress. Such interventions are necessary and are a form of expression of the institutional responsibility of the public university to its servants. Such responsibility, however, needs to be better expressed in actions that aim to reduce the probability of developing symptoms and illness or promoting quality of life at work, now recognized as a necessary strategy to attract and retain qualified personnel suitable for the organizational profile [71]. Although we focus on the UFMG case explored in our research, the results can be understood as a more general alert, in the sense that public policies aimed at mental health prevention cannot ignore working conditions in their variations by occupation.

About the changes to be implemented, which our results support, we recommend actions such as: simplifying and reducing bureaucratic processes, gradually evaluating their needs, rationales, and effectiveness for the proposed objectives; better evaluate/resize the servants’ demands, matching them better with the material conditions and infrastructure in general that make them feasible; deepen analyzes on the persistence of practices of violence and discrimination; strengthen respect for individual limitations; valuing the servant’s work organization strategies, which better balance the various spheres of their personal life; better observe the differences in the attributions of professors and TAS in the adoption and expansion of computerized work systems; develop more support strategies (support) for the servant in carrying out their tasks, including systematically raising support needs; continue valuing the possibilities of updating knowledge, collective participation and the autonomous responsibility of public servants. Based on the results found, all these actions are valid as prevention strategies in the field of mental health and work.

## 5. Conclusions

The present research contributed to reaffirming the multidimensionality of mental health and the approach that derives from social causation [27,29,38,42]. Likewise, it drew attention to the feasibility of including working conditions among the social aspects that contribute to the health-illness process, as well as this inclusion can be developed by covering the complexity of tendencies in the world of work and its dynamic character. Then, these findings represent theoretical contributions to the field of mental health and work and to the comprehension of the impacts of work precariousness. From the theoretical point of view, the results also incentive the psychologist to value the social dimensions of the studied phenomenon (mental health), which is the opposite of traditional ways of exclusively searching for mental causes. The individuals need to strengthen themselves really, although we should not contribute to blaming them for the illness process in which the conditions of life and work environment are implied.

In addition, we understood that our results represent a challenge for public policymakers, if health and working conditions are usually the objects of different bodies in the States. Thus, it reinforces the importance of the notion of intersectionality in vogue, which consists of the development of integrated actions involving several sectors (governmental or not) in the solution of complex and multidimensional problems [72,73]. We understand that the present study is in line with other papers [74,75,76] that strengthen the notion of a link between mental health and work, evidencing that mental health prevention should involve care and better management of working conditions.

We also noted that the present study was carried out in a period in which the country had already signaled the adoption, in the Brazilian higher education context, of more intense policies guided by neoliberal principles. Such measures were aggravated in the 2018–2022 administration, such as the budgetary cuts, seriously compromising the functioning of the country’s federal public universities, either due to the lack of resources or the non-hiring of staff to fill the positions that opened. Additionally, with the outbreak of the COVID-19 pandemic, these servants had to shift their work to a remote format without being properly prepared to conduct their activities or a space reserved in their own residence to carry out their duties, requiring self-funding of equipment and electronic devices, permanent negotiation with family members in the division of household chores, among others. Studies have already shown the impacts of the pandemic on mental health [77,78,79]. Even with the return to face-to-face activities, precariousness at work has intensified, with the new requirements of adaptation to face the proliferation of the virus and having scarce resources. Given this context, we question whether current working conditions have worsened the quality of mental health of public servants, professors, and TAS. Various more specific questions can be elaborated to future research activities, such as: how do the servants perceive the impairment between Demands, Material Precariousness, and Physical Environment after the experience with the referred pandemic and with the problems during the 2018–2022 federal administration? Did they change the perception of organizational support and protection? Were the Respect for Limitations, Knowledge Update, Responsibility, Participation, and Flexibility of Working Hours factors able to protect their mental health? Which profile of mental health was maintained or changed? Was the relationship between mental health profile and the working conditions factor strengthened?

In addition to the moderation of contract and legal conditions, further studies can explore a greater number of aspects, such as salary, the difference between the number of contracted hours and the actual working hours, and greater diversity of work relationships (outsourced, self-employed, etc.). Likewise, they can integrate other moderators that the literature has already explored, such as sleeping [73], working in shifts, and its irregularity [74]. Additionally, we consider it of special importance to introduce more aspects related to remote activities among working conditions, whose adoption accelerated during the COVID-19 pandemic [77,78]. In our research, we had a Computerization factor which, in the current context, is no longer enough. Attention to the dynamic nature of working conditions should not be neglected. We understand that periodic evaluations of the relationship between these two phenomena (working conditions and mental health) are necessary as a basis for public and organizational policies.

## Figures and Tables

**Figure 1 ijerph-20-01536-f001:**
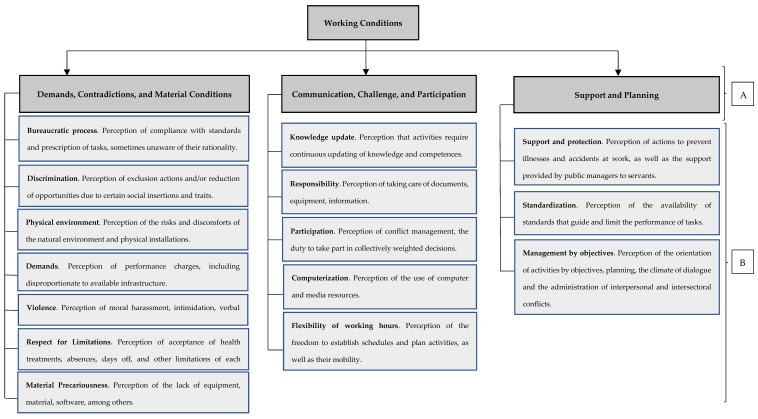
Factorial Structure of First and Second Order regarding the Work Conditions Questionnaire (WCQ). Legend: (**A**) = Factorial Structure of Second Order; (**B**) = Factorial Structure of First Order. Source: Borges, Barbosa, Ansoleaga, Barros, Heleno, and Rentería [51].

**Figure 2 ijerph-20-01536-f002:**
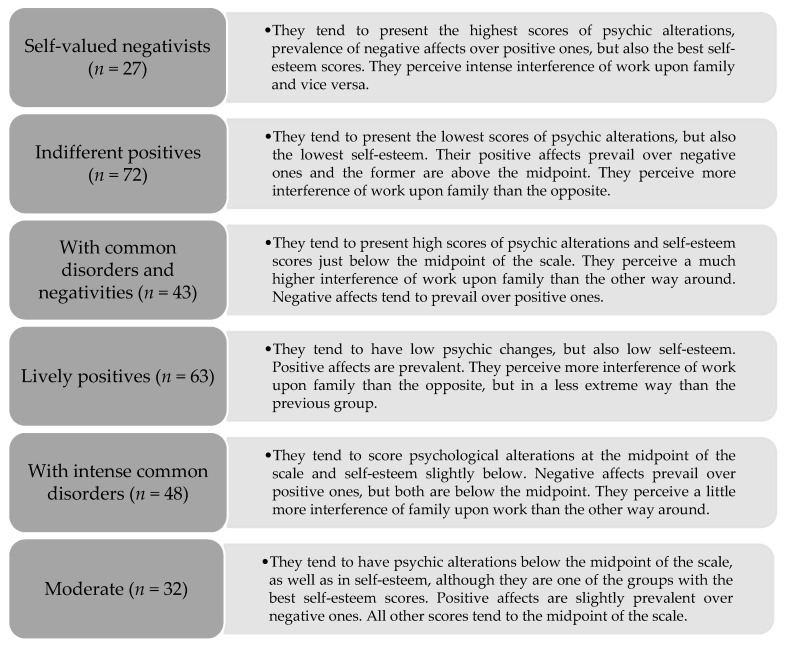
Profiles of mental health.

**Table 1 ijerph-20-01536-t001:** Distribution of scores in the working condition factors (*n* = 285).

Factors of Working Conditions	Mean	Standard Deviation	Percentiles/Quartiles		
			25%	50%	75%
**Demands, Contradictions, and Material Conditions**					
F1 Bureaucratic process	2.58	0.88	1.91	2.56	3.15
F2 Discrimination	1.50	0.55	1.09	1.33	1.72
F3 Physical environment	2.05	0.65	1.54	1.93	2.45
F4 Demands	3.13	0.83	2.56	3.10	3.70
F5 Violence	2.44	0.70	1.91	2.37	2.84
F6 Respect for Limitations	**3.48**	0.55	3.16	3.54	3.84
F7 Material Precariousness	3.31	0.95	2.65	3.35	3.99
**Communication, Challenge, and Participation**					
F1 Knowledge update	3.42	0.77	2.85	3.50	4.00
F2 Responsibility	3.12	0.92	2.54	3.17	3.80
F3 Participation	3.21	0.88	2.64	3.30	3.71
F4 Computerization	**4.27**	0.69	3.94	4.48	4.78
F5 Flexibility of Working hours	2.90	0.64	2.48	2.88	3.35
**Support and Planning**					
F1 Support and Protection	2.45	0.63	2.04	2.40	2.81
F2 Standardization	**3.70**	0.75	3.19	3.74	4.22
F3 Management by Objectives	3.57	0.70	3.14	3.64	4.08

**Table 2 ijerph-20-01536-t002:** Participants’ scores by the indicators of mental health (*n* = 285).

	Mean	Standard Deviation	Percentiles/Quartiles		
			25%	50%	75%
Psychic Changes (GHQ)	2.08	0.71	1.57	2.00	2.57
Positive Affects	2.91	0.91	2.14	2.93	3.64
Negative Affects	2.41	0.75	1.81	2.25	3.00
Self-esteem	2.12	0.49	1.80	2.00	2.40
Interference of Work in Family	3.49	1.39	2.33	3.33	4.67
Interference of Family in Work	2.09	1.12	1.00	1.75	2.75

**Table 3 ijerph-20-01536-t003:** Analysis of the empiric factors of working conditions by mental health profiles.

Working Condition Factors	F Coefficient	Significance Level
Demands, Contradictions and Material Conditions		
F1 Bureaucratic process	10.67	*p* ≤ 0.001
F2 Discrimination	4.89	*p* ≤ 0.001
F3 Physical Environment	6.19	*p* ≤ 0.001
F4 Demands	11.10	*p* ≤ 0.001
F5 Violence	12.34	*p* ≤ 0.001
F6 Respect for Limitations	4.59	*p* ≤ 0.001
F7 Material Precariousness	8.53	*p* ≤ 0.001
Communication, Challenge, and Participation		
F1 Knowledge update	9.79	*p* ≤ 0.001
F2 Responsibility	2.83	*p* = 0.016
F3 Participation	12.19	*p* ≤ 0.001
F4 Computerization	0.94	*p* = 0.457
F5 Flexibility of Working Hours	9.26	*p* ≤ 0.001
Support and Planning		
F1 Support and Protection	8.15	*p* ≤ 0.001
F2 Standardization	1.70	*p* = 0.134
F3 Management by Objectives	10.04	*p* ≤ 0.001

## Data Availability

Not applicable.

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
