# Peer review of "Working Conditions and Mental Health in a Brazilian University"

_ijerph, 2023, doi:10.3390/ijerph20021536_

Round 1
Reviewer 1 Report
The paper provides valuable insight about factors in be considered for improving working conditions in universities to prevent mental illnesses. Anyway, there are some issues that need to be handled, particularly the academic contribution and the coverage of literature review that is not adequately covered. I encourage the authors to consider the comments given below and revise the paper accordingly in order to enhance the overall quality and completeness of the paper.
(1) While the paper provides important benefits from the practical perspective, the contribution in terms of new knowledge is still not adequately clarified. In the introduction, it is important to provide a quick overview of prior studies that have been done on a similar topic and then clearly mention how your research fill some gap that has not been explored when comparing to prior study.
(2) The literature review still has some room to fill. At this point, lack of empirical evidence regarding the workplace factors that affect wellbeing of university employees is included. To strengthen the coverage of literature review, there are some important papers that need to be added as empirical supports for the issues related to factors in a workplace (e.g., workload, formalized structure) that affect stress and wellbeing that university employees generally encounter in recent situations. I have some recommendation for the recent papers that provide solid evidence on this point. I would like the authors to consider including the recommended papers below as the additional references to make the coverage of literature review more completed.
- How Does Mindfulness Help University Employees Cope with Emotional Exhaustion during the COVID-19 Crisis? The Mediating Role of Psychological Hardiness and the Moderating Effect of Workload, Scandinavian Journal of Psychology. 63(5), 449-461 https://doi.org/10.1111/sjop.12826
- Exploring burnout among university faculty: incidence, performance, and demographic issues. Teaching and Teacher Education, 20(7), 713-729. https://doi.org/https://doi.org/10.1016/j.tate.2004.07.002
- Effects of Workplace Rumors and Organizational Formalization During the COVID-19 Pandemic: A Case Study of Universities in the Philippines, Corporate Communications: an International Journal, 26(4), 793-812. https://doi.org/10.1108/CCIJ-09-2020-0127
- Quality of life and burnout among faculty members: How much does the field of knowledge matter?. PLOS ONE 14(3): e0214217. https://doi.org/10.1371/journal.pone.0214217
(3) The authors mentioned that they collected data from civil servants from the Universidade Federal de Minas Gerais. However, it is important to provide some more clarification about this sample group (the nature of the organization and the characteristics of work they perform). A brief explanation about the research context will be helpful for readers to understand how this sample group is appropriate for the objectives of the study.
Author Response
We send detailed answer in attachment file.
Thank you very much for your suggestions.

Reviewer 2 Report
Dear colleagues,
The work presented is of great rigor, and is exceptionally well documented. The topic is relevant, and its exposition is clear. However, there are some aspects that could be improved, and I hope that the authors will find them useful:
- The theoretical background is adequate. However, little emphasis is placed on why a phenomenon as broad as mental health is conceptualized in these variables specifically.
- In the sample, there is a high imbalance in the sample in the proportion of faculty and TAS. Why is this? It would be interesting to justify it, as well as to specify in more detail the type of sampling.
- The methodology section is perhaps the one that requires more attention. The design and analysis carried out are adequate; however, the explanation in the analysis section is too brief.
- The procedure section is missing. It should specify the design, ethical criteria, and other aspects of the development of this research. Much of this information is indicated in other points of the method section, but it would be clearer to structure it in the proposed model.
- Regarding the discussion: given the date of data collection, and the impact that COVID has had in the educational field, it would be useful to go deeper into the way in which the pandemic may have altered some of the issues addressed. There is some mention of the COVID situation, but not much depth.
Congratulations for your work. I hope that my contributions will be an enriching contribution to it.
Kind Regards
Author Response
We sent detailed answers in the the attachment file.
Thank you very much for your comments.

Round 2
Reviewer 1 Report
The authors did a satisfactory work revising the paper according to the comments provided earlier. The quality of the paper is now adequately improved. There is no further comment to the author.